# *GhFAD3-4* Promotes Fiber Cell Elongation and Cell Wall Thickness by Increasing PI and IP_3_ Accumulation in Cotton

**DOI:** 10.3390/plants13111510

**Published:** 2024-05-30

**Authors:** Huiqin Wang, Mengyuan Fan, Yongcui Shen, Hanxuan Zhao, Shuangshuang Weng, Zhen Chen, Guanghui Xiao

**Affiliations:** College of Life Sciences, Shaanxi Normal University, Xi’an 710062, China; wanghuiqin@snnu.edu.cn (H.W.); chenzzzhen@163.com (Z.C.)

**Keywords:** cotton fiber, CRISPR/Cas9, omega-3 fatty acid desaturase, cotton (*Gossypium hirsutum*), phosphatidylinositol

## Abstract

The omega-3 fatty acid desaturase enzyme gene *FAD3* is responsible for converting linoleic acid to linolenic acid in plant fatty acid synthesis. Despite limited knowledge of its role in cotton growth, our study focused on *GhFAD3-4*, a gene within the *FAD3* family, which was found to promote fiber elongation and cell wall thickness in cotton. *GhFAD3-4* was predominantly expressed in elongating fibers, and its suppression led to shorter fibers with reduced cell wall thickness and phosphoinositide (PI) and inositol triphosphate (IP_3_) levels. Transcriptome analysis of *GhFAD3-4* knock-out mutants revealed significant impacts on genes involved in the phosphoinositol signaling pathway. Experimental evidence demonstrated that *GhFAD3-4* positively regulated the expression of the *GhBoGH3B* and *GhPIS* genes, influencing cotton fiber development through the inositol signaling pathway. The application of PI and IP_6_ externally increased fiber length in *GhFAD3-4* knock-out plants, while inhibiting PI led to a reduced fiber length in *GhFAD3-4* overexpressing plants. These findings suggest that *GhFAD3-4* plays a crucial role in enhancing fiber development by promoting PI and IP_3_ biosynthesis, offering the potential for breeding cotton varieties with superior fiber quality.

## 1. Introduction

Cotton, belonging to the Malvaceae family, can be an annual herb or a perennial shrub. Its leaves are broad and ovate, with an acuminate apex and a wide base. The petioles are sparsely pubescent, and the stipules are ovate and sickle-shaped. The flowers are solitary in the leaf axils, with pedicels larger than the petioles. The calyx is slightly short and cup-shaped, and the seeds are oval, covered with long white cotton hairs and gray-white short cotton hairs that are not easily removable. Cotton is used in the production of various textiles for clothing, furniture, and industrial purposes. It is considered one of the most important crops globally due to its high output and low production costs, resulting in relatively inexpensive cotton products. Transgenic cotton refers to cotton varieties that have been genetically modified by introducing beneficial genes from other species into the cotton genome. This genetic modification can lead to the development of high-yielding, high-quality, and insect-resistant cotton strains through scientific research.

Upland cotton (*Gossypium hirsutum*) is a significant fiber crop globally [1]. Cotton fiber, a widely used natural material in textile industries, originates from a specialized single epidermal cell in the seed coat and serves as a valuable model for studying plant cell elongation [2]. The development of cotton fiber involves four main stages: initiation, elongation, secondary cell wall (SCW) biosynthesis, and maturation [3]. The majority of cell wall structures are synthesized in the Golgi apparatus and then transported to the apoplast for cell wall synthesis and remodeling [4,5]. Fiber elongation determines the final length of the mature fiber, while the thickening of the SCW influences the strength and fineness of the mature fiber, which are crucial factors for assessing cotton fiber quality [6]. Cotton fiber development initiates on the ovule surface on the day of anthesis and experiences rapid elongation, typically between 5 and 20 DPA (days post-anthesis), during which the ultimate fiber length is established [7]. Therefore, an increase in fatty acid (FA) synthesis is essential for membrane expansion as fiber cells undergo rapid growth [8].

The biosynthesis of fatty acids occurs in the plastids of plant cells, where the continuous linking of 2-carbon units results in the formation of 16- or 18-carbon long fatty acids that are predominant in the cell membrane. Within the plastids, the soluble fatty acid desaturase can catalyze the conversion of 18:0 to 18:1, with the number 18 representing the carbon atoms in the molecule. The 18:1 fatty acids can then be further transformed into 18:2 either in the plastids or the endoplasmic reticulum (ER). The conversion of 18:1 to 18:2 is facilitated by FAD2 or FAD6 fatty acid desaturation enzymes. FAD2 and FAD6 exhibit similarities in their sequences, although FAD6 possesses a longer N-terminal. The production of 18:2 is triggered by FAD7 or FAD8 fatty acid dehydrogenase, and it can also be converted into 18:3 by the FAD3 enzyme before being exported to the endoplasmic reticulum. FAD7/FAD8 and FAD3 are recognized as omega-3 fatty acid desaturases due to their ability to introduce a double bond at the omega-3 position of the fatty acid structure. Consequently, the enzymes FAD6 and FAD2, responsible for producing 18:2, as well as FAD7/FAD8 and FAD3, which generate 18:3, play a pivotal role in the biosynthesis of polyunsaturated fatty acids (PUFAs) present in all plant species [9].

Fatty acids are crucial components of plant membrane phospholipids and seed storage triacylglycerols. The synthesis of fatty acids plays a significant role in regulating their content, composition, and fiber development [10]. Fatty acid desaturase (FAD) is a key enzyme involved in the production of unsaturated fatty acids by introducing one or more carbon double bonds into hydrocarbon chains at various positions [9]. Specifically, omega-3 fatty acid desaturase (FAD3) is an essential enzyme responsible for converting linoleic acid to linolenic acid (C18:3), which greatly enhances fiber development [10].

Phosphatidylinositol synthase catalyzes the conversion of the substrate C18:3 and CDP-DAG into PI, which is composed of 1,2-DAG phosphate and inositol. PI, a type of membrane lipid, plays a crucial role as a regulatory molecule in the secretion and assembly of cell-wall polymers [11]. The phosphorylation of PI leads to the production of phosphatidylinositol 4-phosphate (PIP4), which can be further converted into PIP2 [12]. In plants, PIPLC, an important lipid hydrolase, cleaves PIP2 to generate two significant secondary messengers, IP_3_ and DAG [13]. Upon phosphorylation, IP_3_ forms inositol hexaphosphate (IP_6_) and can promote fiber cell elongation by increasing ethylene biosynthesis [14,15]. However, the role of FAD3 in cotton remains uncharacterized.

Targeted lipidomics studies reveal that linolenic acid (C18:3) promotes cotton fiber elongation by activating phosphatidylinositol and phosphatidylinositol ponophosphate biosynthesis [10]. GhFAD3 has been found to promote cotton fiber development [10], but its specific mechanism for promoting cotton fiber development has not been confirmed. Secondly, there is no systematic comparison of GhFAD3 family genes. In this study, a *GhFAD3* gene, *GhFAD3-4*, was identified as a promoter of both fiber cell elongation and cell wall thickness. Fibers of over-expressing *GhFAD3-4* showed higher accumulation of PI and IP_3_ compared to wild-type and *GhFAD3-4* knock-out mutants. Silencing GhFAD3-4 resulted in reduced fiber length and cell wall thickness. Additionally, the application of exogenous PI and IP_6_ significantly improved fiber development. These findings suggest that *GhFAD3-4* plays a crucial role in promoting cotton fiber development by enhancing PI and IP_3_ accumulation, leading to increased cell wall deposition and ethylene biosynthesis.

## 2. Results

### 2.1. Genome-Wide Identification of GhFAD3 in Cotton

Previous studies have indicated that linolenic acid C18:3 can enhance cotton fiber development [10,15]. The role of the key dehydrogenase fatty acid FAD3 in linolenic acid synthesis in cotton fiber development remains uncertain. By analyzing AtFAD3, we identified 11 GhFAD3 protein-encoding genes through homology searches in the *Gossypium hirsutum* genome (TM-1 v2.1). Phylogenetic analysis of these GhFAD3 proteins in cotton revealed that GhFAD3-4 and GhFAD3-5 have the closest phylogenetic relationship, while GhFAD3-1 and GhFAD3-2 show a closer phylogenetic connection (Appendix A). Notably, based on database predictions, five *GhFAD3* genes (*GhFAD3-2A*, *GhFAD3-2D*, *GhFAD3-3A*, *GhFAD3-4A*, and *GhFAD3-4D*) exhibit significantly higher expression levels during rapid fiber development stages (Figure 1a). Alignment of multiple sequences unveiled two conserved domains within FAD3 proteins (Appendix A). Furthermore, we examined the expression levels of *GhFAD3* homologous genes at different fiber developmental stages and observed high expression of *GhFAD3-1A* and *GhFAD3-4A* during rapid fiber elongation (Figure 1b–g). Importantly, *GhFAD3-4A and GhFAD3-4D* were notably highly expressed at both 10 DPA and 15 DPA during the rapid fiber elongation period, indicating their potential key role in this stage (Figure 1a,e). In this study, we selected GhFAD3-4D as the research object and abbreviated it as GhFAD3-4.

### 2.2. GhFAD3-4 Positively Regulates Cotton Fiber Development

To investigate the biological function of *GhFAD3-4*, we created transgenic lines with *GhFAD3-4D* overexpression (OE) and knock-out mutants for *GhFAD3-4A/D*, as the coding sequences of *GhFAD3-4A* and *GhFAD3-4D* share a homology of 99.31%. Through pedigree selection and qPCR analysis, we obtained three *GhFAD3-4D-OE* lines and six *GhFAD3-4A/D* knock-out lines (*GhFAD3-4*-KO) using Agrobacterium tumefaciens-mediated transformation of cotton cultivar ‘Jin668’ (Figure 1a–d) [16].

In the *GhFAD3-4* transgenic materials analyzed, the expression levels of *GhFAD3* homologous genes were investigated. *GhFAD3-5A* and *GhFAD3-5D* were notably upregulated in *GhFAD3-4*-OE and downregulated in *GhFAD3-4*-KO. Our study on the *GhFAD3-4* gene’s function may involve the influence of *GhFAD3-5* (Appendix A). *GhFAD3-4* exhibited widespread expression across all tested tissues, with particularly high levels in root and leaf tissues rich in membranous organelles, suggesting a potential key role in membrane expansion (Appendix A). *GhFAD3-4*-OE plants showed significant enhancement in fiber cell length compared to non-transgenic controls, while knocking out *GhFAD3-4A/D* resulted in reduced fiber cell length (Figure 2e,f). Consequently, *GhFAD3-4*-KO plants displayed a marked decrease in cell wall thickness compared to wild-type (WT) and *GhFAD3-4*-OE plants (Figure 2g,h). Fiber twist testing indicated that neither *GhFAD3-4*-OE nor *GhFAD3-4*-KO significantly affected fiber distortion (Appendix A). These findings suggest that *GhFAD3-4* may play a role in promoting cotton fiber elongation and cell wall thickening.

### 2.3. Transcriptome Identification and Characterization of GhFAD3-4 Downstream Genes

To investigate the role of *GhFAD3-4* in cotton fiber development, we conducted transcriptome sequencing to identify differentially expressed genes in *GhFAD3-4* transgenic lines. Our analysis revealed that 1265 and 3458 genes were upregulated in *GhFAD3-4* overexpressing lines compared to control plants and *GhFAD3-4A/D* knock-out lines, respectively (Figure 3a–c). Interestingly, 763 genes were found to be commonly upregulated in both comparisons (Figure 3d). Furthermore, KEGG pathway analysis indicated enrichment of genes involved in inositol phosphate metabolism, with a particular focus on the phosphatidylinositol pathway in linolenic acid regulation (Figure 3e) [17]. 

### 2.4. Expression Analysis of Genes Related to the Inositol Signaling Pathway in GhFAD3-4 Transgenic Materials

Transcriptome analysis of *GhFAD3-4* transgenic material suggested that the phosphatidyl inositol metabolism pathway may be the downstream signaling pathway regulated by *GhFAD3-4* in cotton fiber (Figure 3e). Genes enriched in this pathway include *GhBoGH3B*, *GhMIOX4*, *GhPIS*, and *GhRNF144B*. To investigate their roles in cotton fiber development, we assessed their expression levels at different development stages, including initiation, elongation, secondary cell wall synthesis, and maturation. During the development of cotton fibers from 0 to 20 days, the expression of *GhBoGH3B* gradually increased with time and reached its highest level at 20 days (Figure 4a). *GhMIOX4* showed the highest expression at 0 DPA but was lower at other stages (Figure 4b). *GhPIS* has significantly higher expression during rapid elongation compared to initial and secondary cell wall synthesis (Figure 4c). The expression level of GhRNF144B gradually increased with the passage of time from 0 to 15 days of fiber development and then decreased after 15 days (Figure 4d).

To investigate the regulatory role of *GhFAD3-4* on four specific genes, we analyzed the expression levels of these genes in *GhFAD3-4* transgenic materials (Figure 4e–h). Our results show that *GhBoGH3B* and *GhPIS* expression levels are significantly higher in *GhFAD3-4*-OE compared to the wild type, while *GhFAD3-4*-KO exhibits significantly lower expression levels for both genes (Figure 4e,g). The expression of *GhMIOX4* in *GhFAD3-4* transgenic materials did not show a significant difference (Figure 4f). Interestingly, *GhRNF144B* expression in *GhFAD3-4*-KO materials was significantly higher than in the wild type, with no significant difference observed between *GhFAD3-4*-OE and the wild type (Figure 4h). These findings suggest that *GhFAD3-4* may regulate cotton fiber development by influencing the expression of *GhBoGH3B* and *GhPIS*, key components of the phosphoinositol signaling pathway. Further research is required to confirm the potential involvement of the phosphoinositol signaling pathway in the promotion of cotton fiber development by *GhFAD3-4*.

### 2.5. PI and IP_3_ Can Promote the Development of Cotton Fiber

PI and IP_3_, key components of the inositol signaling pathway, play a crucial role in promoting cotton fiber development by enhancing ethylene biosynthesis [15]. When phosphorylated, IP_3_ transforms into inositol hexaphosphate [18], which possesses diverse functions in plants. The catalytic action of PIPLC on PIP2 leads to the formation of IP_3_, which can subsequently undergo phosphorylation to generate IP_6_ [19]. Our analysis of PI and IP_3_ levels across different developmental stages of cotton fibers revealed peak concentrations at 15 DPA, particularly during the rapid elongation phase of fiber growth (Figure 5a,b). Through an in vitro ovule culture system, we confirmed the positive impact of both PI and IP_3_ on cotton fiber development. Notably, the inhibition of PI resulted in a significant reduction in fiber development, aligning with previous findings (Figure 5b,c,e,f) [15]. 

### 2.6. GhFAD3-4 Promotes Cotton Fiber Development by Increasing PI and IP_3_ Accumulation

To investigate the mechanisms by which *GhFAD3-4* regulates fiber development through phosphatidylinositol, we quantified phosphatidylinositol (PI) content and inositol 1,4,5-trisphosphate (IP_3_) accumulation in the *GhFAD3-4* transgenic lines. Our results showed a significant increase in the levels of PI and IP_3_ in cotton fibers overexpressing *GhFAD3-4* (*GhFAD3-4*-OE), while a decrease was observed in fibers with *GhFAD3-4* knocked out (*GhFAD3-4*-KO) (Figure 6a,b). Subsequently, we utilized an in vitro cotton ovule culture system to validate the role of PI and IP_3_ in *GhFAD3-4*-mediated regulation of fiber development. The experiments demonstrated that *GhFAD3-4*-OE fibers exhibited increased length compared to the wild type, whereas *GhFAD3-4*-KO fibers were shorter (Figure 6c,d). Furthermore, treatment with PI and IP_6_ partially rescued the shortened fiber phenotype in *GhFAD3-4*-KO, while the application of a PI inhibitor significantly inhibited fiber elongation in *GhFAD3-4*-OE (Figure 6c,d). Overall, our findings indicate that the overexpression of *GhFAD3-4* promotes fiber elongation, while the knockout of *GhFAD3-4A/D* hinders this process. Importantly, the in vitro supplementation of PI and IP_6_ successfully reversed the short fiber phenotype induced by *GhFAD3-4A/D* knockout (Figure 6c). These results suggest that *GhFAD3-4* may enhance cotton fiber elongation and thickening of the fiber cell wall by stimulating the accumulation of IP and IP_3_ in cotton fibers.

## 3. Discussion

### 3.1. Various Fatty Acid Synthases Are Involved in the Elongation of Cotton Fibers

Various fatty acids, such as very-long-chain fatty acids (VLCFAs), short-chain fatty acids, and unsaturated fatty acids, have been identified to regulate cotton fiber elongation [20,21,22]. Specifically, two ketoacyl-CoA synthase genes (KCS) are involved in the biosynthesis of VLCFAs, which is crucial for high-grade cotton fiber development [22]. Silencing experiments targeting fatty acid desaturases (GhFAD), PI synthase (PIS), and PI kinase (PIK) resulted in reduced mRNA levels of these genes in fibers, leading to a 10 mm decrease in fiber length compared to control plants [10]. Furthermore, our study highlights the role of unsaturated fatty acid synthase GhFAD3-4 in promoting cotton fiber elongation through the production of C18:3 from C18:2, expanding our understanding of the regulatory mechanisms involved in cotton fiber development [15].

### 3.2. Biological Function of FAD3 Homologs in Plants

Numerous studies have demonstrated the role of FAD3 in promoting the biosynthesis of linolenic acid [23,24,25]. Physaria fendleri FAD3-1 overexpression increases ɑ-linolenic acid content in camelina seeds [23]. Some studies have even conducted molecular-level analyses to determine the specific mechanisms through which linolenic acid operates. Various plant species, including B. napus, Camelina sativa, Linum usitatissimum, Vernicia fordii, Gossypium hirsutum, S. hispanica, Cannabis sativa, and P. frutescens, exhibit a retention signal KXKXX/XKXX at their c-terminus, a key characteristic of FADs. Furthermore, research indicates that BnFAD3 acts as a transmembrane protein, converting omega-6 to omega-3 fatty acids while potentially functioning as a potassium ion channel in the endoplasmic reticulum [24]. In addition, some studies on flaxseed (Linum usitatissimum) have identified the FAD3 protein as a potential source of peptides with angiotensin-converting enzyme (ACE) inhibitory and dipeptidyl peptidase-IV (DPP-IV) activities [25]. This particular study delved into the downstream signaling pathway of GhFAD3-4 and confirmed its impact on cotton fiber development by influencing the accumulation of PI and IP3. The lack of a molecular-level explanation for this phenomenon could potentially pave the way for further in-depth exploration of the functions of FAD3-4 in future research.

### 3.3. The Role of the Phosphoinositol Signaling Pathway in Cotton Fiber Development

Inositol-1,4,5-trisphosphate (IP_3_) is a crucial second messenger produced through the hydrolysis of phosphatidylinositol (4,5) bisphosphate (PIP2) by phosphoinositide-specific phospholipase C (PIPLC). GhPIPLC2D, predominantly expressed in elongating fibers, plays a key role in fiber elongation. A reduction in GhPIPLC2D transcription led to shorter fibers with decreased levels of IP_3_ and ethylene production. Treatment with linolenic acid (C18:3) and phosphatidylinositol (PI), a precursor of IP_3_, enhanced the accumulation of IP_3_, myo-inositol-1,2,3,4,5,6-hexakisphosphate (IP_6_), and ethylene biosynthesis. Silencing GhPIPLC2D resulted in reduced fiber length, which could be rescued by the exogenous application of IP_6_ and ethylene. The positive regulation of fiber elongation by GhPIPLC2D and IP_3_ is linked to enhanced ethylene biosynthesis [15]. Additionally, external application of linolenic acid (C18:3), soybean L-alpha-PI, and specific PIPs containing PIP 34:3 significantly promoted fiber growth, while a liver PI lacking the C18:3 moiety, linoleic acid, and PIP 36:2 were found to be ineffective [10].

The study systematically analyzed the levels of PI and IP_3_ in the inositol phosphate signaling pathway at various stages of cotton fiber development. The function of these molecules was assessed through in vitro ovule culture experiments (Figure 5). The research determined that *GhFAD3-4* promotes cotton fiber development by increasing the accumulation of PI and IP_3_ within the fibers (Figure 6). Furthermore, the study investigated the regulatory effects of *GhFAD3-4* on the expression levels of four genes in the inositol phosphate signaling pathway, revealing significant differences in the expression levels of *GhBoGH3B* and *GhPIS* (Figure 4). These findings suggest that *GhFAD3-4* may influence cotton fiber development by upregulating the expression of these two genes. However, additional experimental evidence is required to further support these conclusions.

## 4. Conclusions

Fiber quality and yield, determined by fiber length and strength, are crucial traits for the textile industry. Our study found that *GhFAD3-4* plays a role in regulating fiber quality by increasing PI accumulation, which is essential for plant cell wall assembly and may enhance fiber cell wall thickness. Additionally, the results showed a significant increase in IP_3_ content in *GhFAD3-4*-OE lines compared to control and *GhFAD3-4* knockout lines, indicating a close relationship to PI decomposition. Therefore, PI and IP_3_ are key regulators of *GhFAD3-4* function. Overall, our study enhances the toolkit for improving agronomic traits (quality and yield) through genetic manipulation in the future.

## 5. Materials and Methods

### 5.1. Plant Material and Growth Conditions

The experimental wild-type upland cotton and transgenic cotton lines were cultivated in a greenhouse under controlled conditions of 28 °C with 14 h of light and 10 h of darkness. Ovules from the same-day flowering were collected for in vitro ovule culture experiments post-flowering, and fiber length was measured upon full maturation of cotton bolls. Cotton fibers were sampled at 5, 10, 15, 20, and 25 days after flowering to analyze the gene expression levels at various stages of fiber development.

### 5.2. Cotton Transformation

The vector construction primers for constructing *GhFAD3-4* overexpression and gene knockout materials are shown in Appendix A. For *GhFAD3-4* genetic transformation, the constructed plasmids were introduced into the LB4404 *Agrobacterium tumefaciens* strain to be used on ‘Jin668’ cotton seedlings. 5–7 mm hypocotyls from the seedlings were incubated with the *A. tumefaciens* suspension (OD600 = 0.1–0.5) for 20 min and then transferred onto a callus induction medium after being blotted dry on sterilized filter papers. After callus induction, proliferation, embryogenic callus induction, embryo differentiation, and plantlet regeneration, the putative transgenic plants were transferred to pots and grown in a greenhouse at 28 °C under a long-day (14 h light/10 h dark) cycle (Figure 7).

Sterile seedling germination medium: 1/2 MS macronutrients, supplemented with 15 g/L glucose and 2.5 g/L Phytagel (Sigma, St. Louis, MI, USA).

Callus induction, proliferation medium: MS inorganic salts, B5 vitamins as the basic medium (hereinafter referred to as MSB), plus different types of hormone combinations.

Somatic embryo differentiation medium: MSB basic medium (KNO_3_ doubled, NH_4_NO_3_ removed) and supplemented with GIn 2.0 g/L and Asn 1.0 g/L.

Embryo germination and rooting medium: 1/2 MS inorganic salts + B_5_ organic matter, addition of 15 g/L glucose and 2.5 g/L phytagel.

### 5.3. Cotton Ovule Culture

The operation method of the cotton ovule culture experiment is as described in the previous literature [15]. One day after fertilization, cotton bolls were collected, and the isolated ovule was removed for the ovule culture experiment. The ovules were first sterilized in a 10% sodium hypochlorite solution, peeled out in a sterile environment, and cultured separately on a fluid nutrient medium with 5 μM PI, 5 μM IP_6_, and 1 μM PI inhibitor (5-hydroxytryptamine) for 15 days.

### 5.4. RNA Extraction and RT-qPCR Analysis

Total RNA was extracted from cotton fibers, roots, stems, and leaves using an RNA isolation kit (Tiangen, Beijing, China) following the manufacturer’s instructions. First-strand cDNA was then reverse-transcribed from the total RNA using a kit from Takara (San Jose, CA, USA). RT-qPCR was conducted with gene-specific primers detailed in Appendix A, with GhUBQ7 (GenBank No. AY189972) serving as the internal control. The reactions were carried out using the Roche LightCycler^®^ 480 II instrument (Roche Holding Ltd., Basel, Switzerland). Relative expression levels of the target genes were calculated using the 2^−ΔΔCT^ method. Each gene in the RT-qPCR analysis underwent three independent biological replications.

### 5.5. RNA-Seq Analyses

Total RNA was extracted from 10-day-old control fibers (CK), *GhFAD3-4*-OE, *GhFAD3-4*-KO. The detailed RNA-seq experiment was as described previously [22]. The differential expression analysis was performed by DESeq2 software (ver. 3.19) between two different groups [26]. The genes with the parameter of false discovery rate (FDR) below 0.05 and an absolute fold change of ≥2 were considered differentially expressed genes (DEGs). The bioinformatic analysis was performed using the Omicshare tools “https://www.omicshare.com/ (accessed on 15 May 2023)”. RNA-seq data was available at the NCBI under accession number PRJNA1084215.

### 5.6. The Natural Twist of Fibers Analysis

To analyze the natural twist of fibers, dried mature cotton fibers were coated with gold using an ion-sputtering machine. Subsequently, a scanning electron microscope (Hitachi, Tokyo, Japan) was employed to image the fibers following the standard procedure outlined in the instruction manual, as reported in previous literature [27].

### 5.7. Microscopic Analysis

Mature fibers from *GhFAD3-4* transgenic and wild-type cotton plants were embedded in optimum cutting temperature compound (O.C.T. Compound, CAS No. 4583, Solarbio, Beijing, China) for 30 min at −20 °C. The samples were then sliced into 5-μm-thick sections by microtome CryoStar™ NX70 (ThermoFisher Scientific, Waltham, MA, USA), and the sections were dried at 65 °C for 30 min. The dried samples were then fixed in absolute ethanol at room temperature for 5 min, and the cell wall sections were stained with 0.01% (*w*/*v*) calcofluor white (Sigma, St. Louis, MI, USA) at room temperature for 5 min. The stained cross sections of fiber cells were observed at 405 nm and 100× under a confocal laser scanning microscope TCS SP8 STED (Leica, Solms, Germany).

## Figures and Tables

**Figure 1 plants-13-01510-f001:**
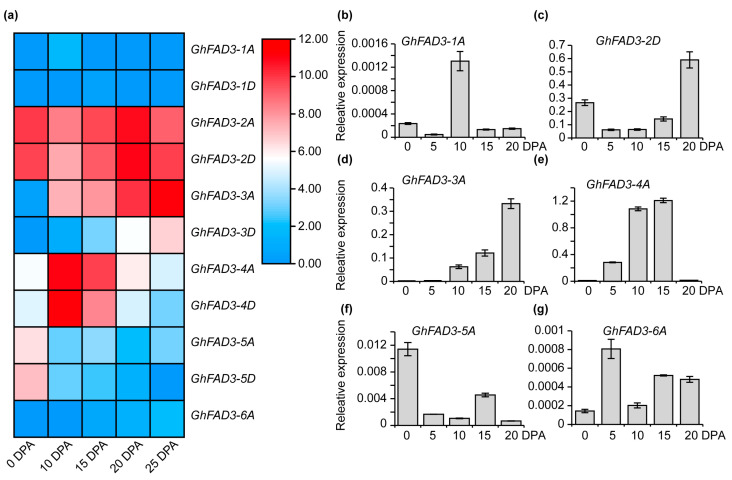
Genome-wide identification of *GhFAD3* in cotton. (**a**) Heatmap of *GhFAD3* expression at five stages of fiber development. (**b**–**g**) Relative expression levels of *GhFAD3-4* homologs in fibers at different developmental stages. DPA, day-post anthesis.

**Figure 2 plants-13-01510-f002:**
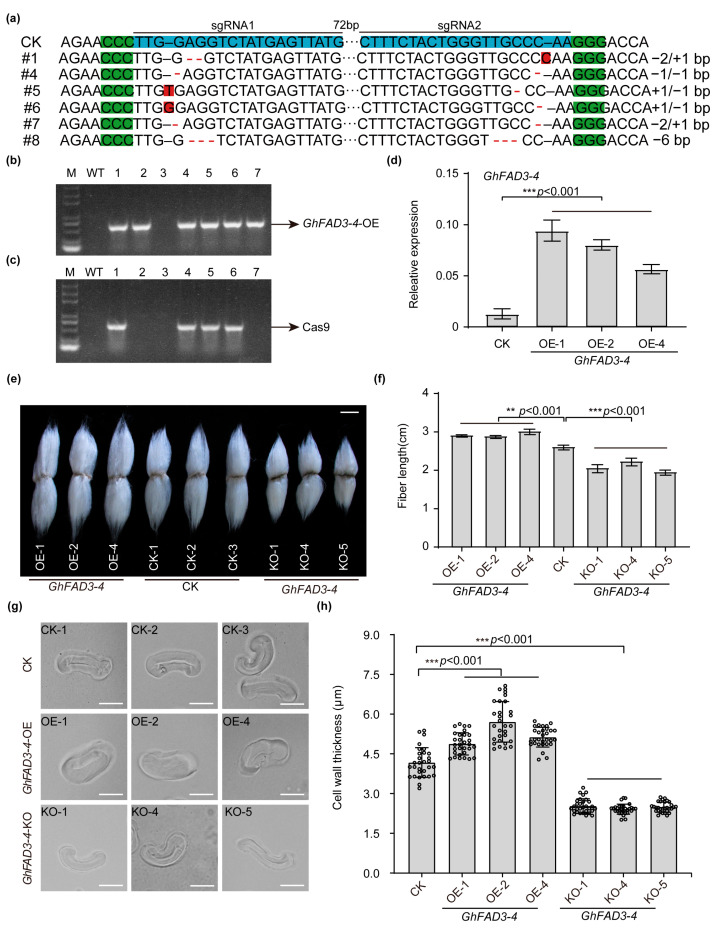
*GhFAD3-4* positively regulates cotton fiber development. (**a**) Sanger sequencing-based genotyping of *GhFAD3-4*-knockout lines obtained by CRISPR-Cas9. Nucleotide deletions are indicated by the red dashes. (**b**,**c**), Identification of the *GhFAD3-4* overexpression (**b**) or knockout lines (**c**) by PCR. M: marker; WT: wild-type; OE: overexpression; KO: knockout. (**d**) Identification of the *GhFAD3-4* overexpression lines by PCR. (**e**,**f**), Phenotypes (**e**) and length measurement (**f**) of mature fibers from *GhFAD3-4*-OE, *GhFAD3-4*-KO, and control plants. Scale bar = 1 cm. (**g**) Cotton fiber cell wall thickness phenotype from WT, *GhFAD3-4*-OE, *GhFAD3-4*-KO. Bars = 20 μm. (**h**) Mean cell wall thickness of fibers from *GhFAD3-4*-OE, *GhFAD3-4*-KO, and control cotton plants. Thirty fibers from three ovules were used for each sample, and each fiber was measured three times. OE: overexpression. CK: control. KO: knock-out. *** and ** indicate *p* < 0.001 and *p* < 0.01 (Student’s *t* test).

**Figure 3 plants-13-01510-f003:**
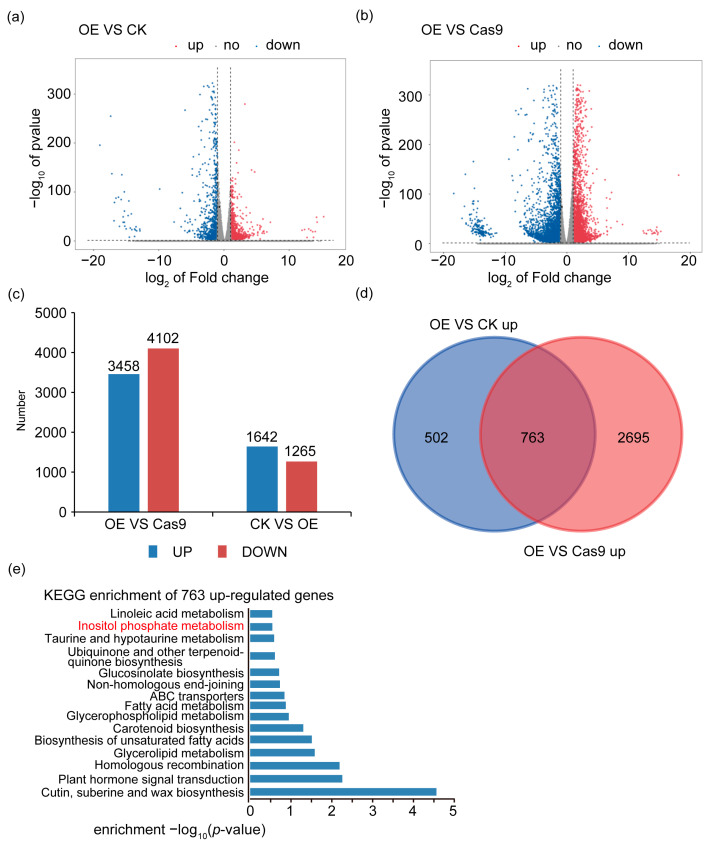
Transcriptome identification and characterization of *GhFAD3-4* downstream genes. (**a**) Volcanic maps of differentially expressing genes in *GhFAD3-4*-OE lines compared with the control. Blue dots: significantly upregulated genes; red dots: significantly downregulated genes; Grey dots, non-differentially expressed genes. (**b**) Volcanic maps of differentially expressing genes in *GhFAD3-4*-OE lines compared with *GhFAD3-4* knockout lines. Blue dots: significantly upregulated genes. Red dots: significantly downregulated genes. Grey dots: nondifferentially expressed genes. (**c**) The bar graph counts the number of differentially expressing genes in the *GhFAD3-4*-OE lines compared with control and *GhFAD3-4* knockout lines. (**d**) Venn diagram showing the 763 overlapping genes in the DEGs that were upregulated in the *GhFAD3-4*-OE lines compared with control and *GhFAD3-4* knockout lines, respectively. DEGs: differentially expressed genes. (**e**) KEGG enrichment analysis of the 763 overlapping upregulated DEGs.

**Figure 4 plants-13-01510-f004:**
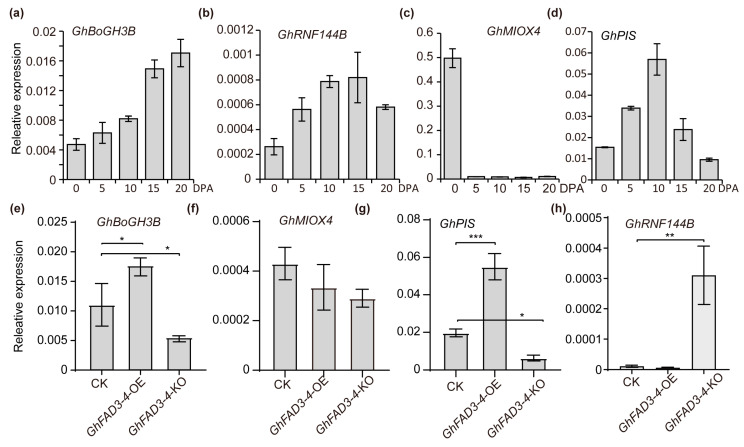
The expression of genes related to the phosphoinositol signaling pathway in *GhFAD3-4* transgenic materials. (**a**,**c**,**e**,**g**) are the expression levels of the *GhBoGH3B*, *GhMIOX4*, *GhPIS,* and *GhRNF144B* genes at different stages of fiber development, respectively. (**b**,**d**,**f**,**h**) are the expression levels of the *GhBoGH3B*, *GhMIOX4*, *GhPIS,* and *GhRNF144B* genes in *GhFAD3-4* transgenic materials, respectively. ***, ** and * indicate *p* < 0.001, *p* < 0.01 and *p* < 0.05 (Student’s *t* test).

**Figure 5 plants-13-01510-f005:**
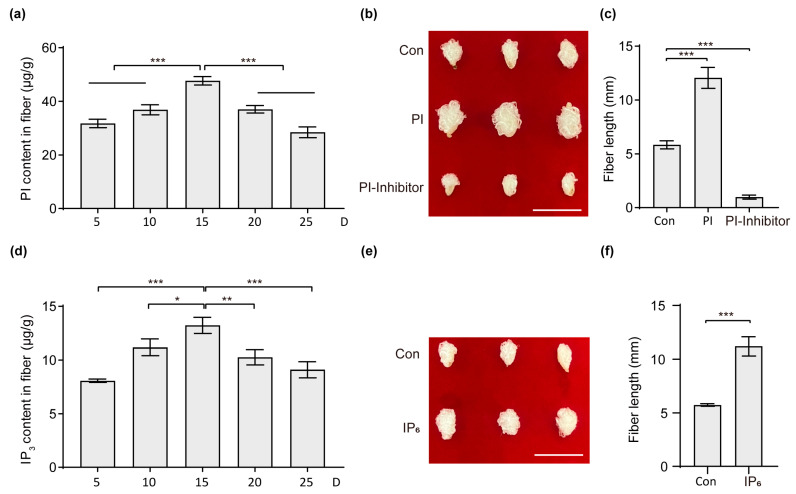
Effect analysis of PI and IP3 on cotton fiber development. (**a**,**d**), PI (**a**) and IP3 (**d**) contents were detected at different stages of fiber development. (**b**,**e**) Phenotypic analysis of effects of PI (**b**) and IP3 (**e**) on ovule fiber development cultured on medium supplemented with 5 μM PI, 5 μM IP6, and 1 μM PI inhibitor (5-hydroxytryptamine) for 15 days. Scale bars = 0.5 cm. Statistical significance for each comparison is indicated (Student’s *t*-test): *, *p* ≤ 0.05, **, *p* ≤ 0.01, ***, *p* ≤ 0.001. (**c**,**f**) are the statistical results of the fiber length of (**b**,**e**), respectively.

**Figure 6 plants-13-01510-f006:**
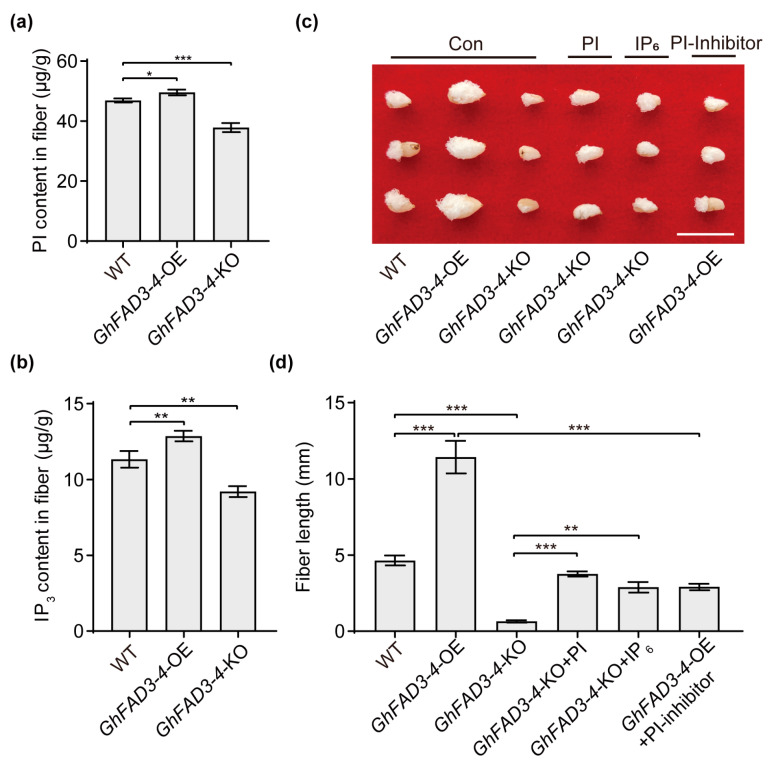
*GhFAD3-4* promotes cotton fiber growth by increasing PI and IP_3_ accumulation. PI (**a**) and IP_3_ (**b**) content in fibers from *GhFAD3-4*-OE, *GhFAD3-4*-KO, and control plants. (**c**) Phenotypes of fibers from the *GhFAD3-4*-OE, *GhFAD3-4*-KO, and control lines cultured on medium supplemented with 5 μM PI, 5 μM IP_6_, and 1 μM PI inhibitor (5-hydroxytryptamine) for 15 days. Scale bars = 1 cm. Statistical significance for each comparison is indicated (Student’s *t*-test): *, *p* ≤ 0.05, **, *p* ≤ 0.01, ***, *p* ≤ 0.001. OE: overexpression. CK: control. KO: knockout. (**d**) is the statistical analysis of fiber length in (**c**).

**Figure 7 plants-13-01510-f007:**
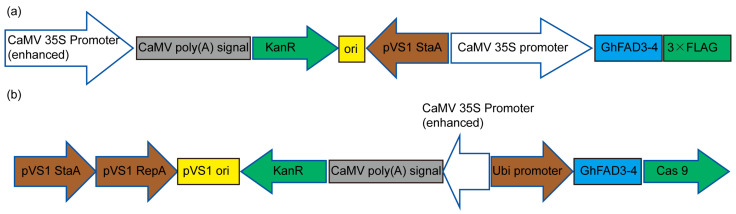
*GhFAD3-4* overexpression and gene knockout material vector map. (**a**) Vector maps of *GhFAD3-4* overexpression. (**b**) Vector maps of *GhFAD3-4* gene knockout.

## Data Availability

All relevant data are within the manuscript and its Appendix A.

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
