# Peer review of "GhFAD3-4 Promotes Fiber Cell Elongation and Cell Wall Thickness by Increasing PI and IP3 Accumulation in Cotton"

_plants, 2024, doi:10.3390/plants13111510_

Round 1
Reviewer 1 Report
Comments and Suggestions for Authors
GhFAD3-4 promotes fiber cell elongation and cell wall thick- 1 ness by increasing PI and IP3 accumulation in cotton
Abstract:
Explain with different parameters like cotton fiber, roots and leaves etc.
Introduction
Write about some sentences on wild cotton, nature, production, features and also about trans genic.
Results and discussions
Line #150: 2.4. Expression analysis of genes related to inositol signaling pathway in GhFAD3-4 transgenic
Please explain each stage giving the name of stage, separately transgenic and control. different graphs.
How about morphological parameters, yield etc. in both cotton?
How about cell wall thickness, is not morphological measurable? micrometers etc.
Materials and methods
Line # 319: 5.4. RNA extraction and RT-qPCR analysis
Total RNA was extracted from cotton fibers, roots, stems, and leaves, dose they were same, you mix them, definitely, cotton fiber RNA will different from others and same for the others, explain in different parameters.
Reference:
Make references accordingly. according to journal.
Reviewer 2 Report
Comments and Suggestions for Authors
This study by Wang et al. aims to investigate the role of GhFAD3-4 enzymes in the development of cotton fibre by producing OE and knockout lines for GhFAD3-4 and performed transcriptome sequencing and characterization of the transgenic plants. I think overall the experiments are planned meticulously. However, the main criticism that I have is that the article does not provide a clear background on what is known in the field. Other studies have done similar work, and the idea is not novel; the authors have gone a step further and have picked up more genes through phylogenetic analysis, expression studies, and have shown the physiological data and highlighted the role of this enzyme in fibre elongation and cell wall thickening by carefully planning the experiments. I think they need to clearly state what has been known before and what they have done.
I have found missing references at multiple places, such as “PI and IP3, key components of the inositol signaling pathway, play a crucial role in promoting cotton fiber development by enhancing ethylene biosynthesis” needs a reference; otherwise, it looks like the authors have uncovered this role in the current study.
More importantly, the authors need to be consistent and clearer about the numbering of FAD3 homologues. For example, in line 90 they used GhFAD3-4A then in line 91 they used GhFAD3-4; this is very confusing, and this (Figure 1a, e) is Figure 1b-e? Similarly, GhFAD3-4A/D, is a double mutant? although they share >99% similar sequence, they are independent genes so it must be stated as such.
It would be interesting to know the reason for not choosing GhFAD3-2A, 2D, and 3A for the downstream analysis.
For the discussion, section 3.2 is all over the place without sending a clear message; the authors need to work on it too.
Comments on the Quality of English Languagegood
Reviewer 3 Report
Comments and Suggestions for Authors
Reviewer comments:
The manuscript ID plants-2985854 entitled “GhFAD3-4 promotes fiber cell elongation and cell wall thick ness by increasing PI and IP3 accumulation in cotton” I found this research topic is encouraging, the study focused on GhFAD3-4 gene promote fiber cell elongation and cell wall thickness by increasing PI and IP3 accumulation in cotton. But I have few concerns related to the research article. I am asking authors to revise the manuscript carefully considering my comments for possible publication in “Plants”.
Abstract: Good.
Introduction: Good.
Materials and Methods:
• Line no 303: The authors requested to add vector details and vector maps used in GhFAD3-4 overexpression and gene knockout.
• Authors are requested to add gRNA sequence in Supplementary Table S1.
• Line no 305: The authors suggested to check and correct “LB4404”
• Authors are requested to add media composition for callus induction, proliferation, embryogenic callus induction, embryo differentiation, and plantlet regeneration.
• Authors are requested to add genome wide identification of FAD3-4, transcriptome analysis.
Results:
• The authors suggested replacing PCR images, image quality is not good Marker is not visible.
The submitted manuscript may be acceptable for publication after a minor revision.
